# Stomach Cancer and Exposure to Talc Powder without Asbestos via Chinese Herbal Medicine: A Population-Based Cohort Study

**DOI:** 10.3390/ijerph16050717

**Published:** 2019-02-28

**Authors:** Che-Jui Chang, Yao-Hsu Yang, Pau-Chung Chen, Hsin-Yi Peng, Yi-Chia Lu, Sheng-Rong Song, Hsiao-Yu Yang

**Affiliations:** 1Institute of Environmental and Occupational Health Sciences, National Taiwan University College of Public Health, Taipei 100, Taiwan; B95401101@g.ntu.edu.tw(C.-J.C.); pchen@ntu.edu.tw (P.-C.C.); yope159@gmail.com (H.-Y.P.); 2Department of Family Medicine, National Taiwan University Hospital, Taipei 100, Taiwan; 3Department of Traditional Chinese Medicine, Chiayi Chang Gung Memorial Hospital, Chiayi 613, Taiwan; r95841012@ntu.edu.tw; 4Health Information and Epidemiology Laboratory of Chang Gung Memorial Hospital, Chiayi 613, Taiwan; 5School of Traditional Chinese Medicine, College of Medicine, Chang Gung University, Taoyuan 333, Taiwan; 6Department of Public Health, National Taiwan University College of Public Health, Taipei 100, Taiwan; 7National Taiwan University Hospital and National Taiwan University College of Medicine, National Taiwan University Hospital, Taipei 100, Taiwan; 8Office of Occupational Safety and Health, National Taiwan University College of Medicine, Taipei 100, Taiwan; 9Department of Geosciences, National Taiwan University, Taipei 106, Taiwan; d00224007@ntu.edu.tw (Y.-C.L.); srsong@ntu.edu.tw (S.-R.S.)

**Keywords:** asbestos, talc, gastric cancer, herbal medicine, drug safety

## Abstract

The present investigation was designed to explore the risk of stomach cancer by oral intake of talc powder without asbestos. We conducted a population-based cohort study on a randomly sampled cohort from Taiwan’s health insurance database, with population of 1,000,000. The study participants were followed up through 2013. The outcome event of interest was the diagnosis of stomach cancer. The exposure of interest was the prescription of talc powder. Cox regression analyses were performed respectively. There were 584,077 persons without talc exposure and 21,575 talc users, 1849 diagnosed with stomach cancer. Persons with exposure of talc had a higher hazard ratio of stomach cancer (adjusted hazard ratio, 2.13; 95% confidence interval (CI), 1.54–2.94; *p* < 0.001). Classification by cumulative exposure of talc yielded adjusted hazard ratios of stomach cancer of 1.58 (95% CI, 0.79–3.17; *p* = 0.19) and 2.30 (95% CI, 1.48–3.57; *p* < 0.001) among persons with high (>21 g) and medium (6–21 g) exposure of talc, as compared to the low-exposure counterparts. Our data demonstrated positive association between increased risk of stomach cancer and oral intake of talc without asbestos. Despite the absence of dose-response effect, there might be a link between stomach cancer and talc.

## 1. Introduction

Talc is a mineral widely used in various industries, including food and cosmetics. In Chinese herbal medicine, talc is used as an antipyretic and diuretic agent [1]. The safety of talc powder ingested by or applied to humans is therefore important. The association of stomach cancer and talc was first discussed in the 1970s, because of high incidence of stomach cancer and treatment of rice with talc in Japan [2]. However, regarding the toxicity and carcinogenicity of talc, we must distinguish between talc with and without asbestos. There was possible contamination of talc by asbestos in that research, and the hypothesis was not confirmed by further review of case-control studies [3]. Therefore, it is essential for researchers to focus on the safety of talc without asbestos.

The carcinogenicity of talc in human has been studied and well-documented. The International Agency for Research on Cancer (IARC) classifies talc containing asbestos as “carcinogenic to humans” (group 1), while inhaled talc without asbestos is considered “not classifiable as to carcinogenicity in humans” (group 3) [4]. Perineal talc application is listed by the IARC as possibly carcinogenic to humans (group 2B). For ingestion of talc without asbestos, however, recent studies have not reached conclusion on its carcinogenicity in human gastrointestinal system. Accordingly, we need more evidence to clarify this issue. The objective of this study is to explore the risk of stomach cancer by oral intake of talc without asbestos. We hypothesized that a positive correlation exists with a dose-response relationship.

## 2. Materials and Methods

We conducted a population-based cohort study using the National Health Insurance Research Database (NHIRD) of Taiwan, comparing the incidence of stomach cancer among people with different levels of oral talc intake. We followed the Strengthening the Reporting of Observational Studies in Epidemiology (STROBE) statement checklist for cohort studies, version 4 [5].

### 2.1. Setting

This study used data from the Longitudinal Health Insurance Database 2005 (LHID 2005) to assess population with exposure of talc from Chinese herbal medicine and the risk of stomach cancer. In Taiwan’s NHIRD, which enrolled 99.6% of Taiwan’s population [6], there are complete drug prescription files and original claim data for reimbursement, including the prescription data of talcum powder. LHID 2005 contains all original claim data of 1,000,000 beneficiaries enrolled in 2005 stratified randomly sampled from the 2005 Registry for Beneficiaries of the NHIRD [7]. The datasets in the LHID 2005 used for this study were as follows:Registry for Beneficiaries contains the registration data of all beneficiaries.Registry for drug prescriptions contains the names of all approved medications and related information including the drug codes issued by the National Health Insurance Administration (NHI)Registry for Catastrophic Illness Patients contains the profiles of patients in the catastrophic illness program, which waives co-payments for expenses related to the disease that qualifies patients to the program [8]. Cancers are included in the catastrophic illness program [9].Ambulatory Care Expenditures by Visits contains the physician billing claims data, including the diagnosis, the medical orders and the names of medication prescribed.Details of Ambulatory Care Orders connects the Ambulatory Care Expenditures by Visits to show the details of drug prescription record, including the days of prescription, the frequency of usage, and the total amount prescribed by a physician.

We set the study period started from 1 January 1997, consistent with the first enrollment year of the NHIRD, and ended on 31 December 2013.

### 2.2. Participants

We collected the original claim data from 1,000,000 beneficiaries randomly sampled in stratification on the basis of demographic information from the 2005 Registry for Beneficiaries of the NHIRD [7]. The exclusion criteria were: patients younger than 20 on 1 January 1997,patients with diagnosis of cancer in or before 1997,patients with gastric ulcer, duodenal ulcer, peptic ulcer, gastritis, duodenitis, or *Helicobacter pylori* infection in or before 1997

### 2.3. Variables

The outcome event of interest was the diagnosis of stomach cancer, confirmed by the Registry for Catastrophic Illness Patients with International Classification of Diseases, Ninth Revision, Clinical Modification (ICD-9-CM) code 151. The start date of all included subjects was 1 January 1997. The end date of follow-up was the date of confirmed diagnosis of stomach cancer, drop-out of the insurance program, or 31 December 2013, whichever came first. The exposure of interest was the prescription of talc powder, queried by the drug codes issued by the NHI of Taiwan. Factors considered potential confounders were age, gender, monthly income, region of residence, and co-morbidities. The monthly income and the region of residence were based on the data of Registry for Beneficiaries when joining the NHI program. Co-morbidities were assessed using the Charlson Comorbidity Index (CCI) [10] in 1997 with exclusion of cancer, as patient with cancer were already excluded from the study. We defined the diagnosis of the co-morbidities as receiving no less than three times of same diagnosis within one year based on the ICD-9-CM codes from the physician billing claims data in the LHID 2005. We calculated the CCI using the clinical conditions and associated scores as follows:

1 point each: acute myocardial infarction or history of myocardial infarction, congestive heart failure, peripheral vascular disease, dementia, cerebrovascular disease, chronic lung disease, connective tissue disease, ulcer, chronic liver disease, diabetes without complications.

2 points each: Hemiplegia, moderate or severe kidney disease, diabetes with end organ damage.

3 points each: Moderate or severe liver disease.

6 points each: Acquired immune deficiency syndrome.

The technical part of CCI calculation was based on the open-sourced Statistical Analysis System (SAS) scripts published by the Health Care Delivery Research at the National Cancer Institute of the United States [11]. We also open-sourced the SAS scripts of this study in the Appendix A.

We converted the quantitative variables into categorical variables as follows. Elderly was defined as age of 65 years or greater, which is consistent with a general definition [12]. Region codes of the residential region were transformed into three levels of urbanization according to Taiwan NHRI publications, with level 1 referring to the “most urbanized” and level 3 referring to the “least urbanized” communities [13]. Increased comorbidity score was defined as CCI of 3 or greater, which was adopted or suggested by previous studies [14,15]. Regarding the talc exposure, we used the first and the third quartile of cumulative dose in subjects with talc intake as cut-off points to categorize the exposure into three levels.

### 2.4. Exposure Assessment

The amount of oral administration of talc was accumulated since the first time of prescription using the drug prescription record in the LHID 2005. NHI of Taiwan only reimburses prescription of talcum powder in the form of extracted Chinese herbal products; Therefore, talc in the form of traditional Chinese herbal decoction was not included in the database. We treated the talc exposure as a time-dependent variable in order to eliminate the immortal time bias, which is a form of selection bias arising when the period between cohort entry and date of first exposure to a drug is either misclassified or simply excluded because the event of interest has not occurred [16]. For patients ever received medical prescription of talc, we considered the time interval between the beginning of the study (1 January 1997) and the date of first prescription of talc to be a non-exposure period, whereas the time interval from the date of first prescription of talc to the endpoint of follow-up was recognized as an exposure period.

### 2.5. Statistical Analysis

Multivariate Cox proportional hazards regression was used to calculate the hazard ratios (HR) and 95% confidence intervals (CI) for the association between talc exposure and risk of stomach cancer. Age (in categories), gender, and potential confounders listed above were included in the model. Patients were censored if they were lost to follow-up.

A sensitivity analysis is often needed to determine how different settings of parameters influences the prediction model under a given set of assumptions. In the original model, we used the cut-off points from the first and the third quartile of all values of cumulative talc exposure in subjects with talc intake to categorize the cumulative talc exposure into three levels (≤first quartile or unexposed, first quartile–third quartile, >third quartile). To minimize the impact of arbitrariness, we applied zero and the median value of the cumulative talc exposure as new cut-off points, in order to make three levels of talc exposure (unexposed, ≤median, >median) in the sensitivity analysis. We also applied a minimal induction period for stomach cancer by excluding observations with time to event less than five year. This exclusion was independent to the selection of talc-exposed subject and could make sure all cases of stomach cancer with talc exposure would have at least five years of exposure.

A two-tailed *p* value of 0.05 was considered to be statistically significant. All statistical analyses were conducted using SAS software (version 9.4; SAS Institute, Cary, NC, USA).

## 3. Results

Figure 1 summarizes the data collection diagram of the study. Of the 1,000,000 subjects in the LHID 2005, there were 605,690 after excluding subjects with age below 20 and patients diagnosed with cancer in or before 1997, which is the first year of the study. With 38 dropping out from the NHI program at the beginning, 605,652 subjects remained, with 9,909,104 people years of follow-up and an average follow-up period of 16.4 years. There were 584,077 (96.4%) subjects without talc exposure and 21,575 (3.4%) talc users. Followed through 31 December 2013, there were 1849 (0.3%) subjects diagnosed with stomach cancer (the event). Figure 2 describes the reasons for prescription of talcum powder. The diagnoses accounting for more than 3% of all talc prescription were shown in the bar chart with percentages. As shown in our data, talc was prescribed mainly in infectious diseases for symptom relief.

The flow diagram reports the numbers of individuals at each stage of the study.

The bar chart shows the ICD codes and the percentages of all prescription of talcum powder. ICD codes of which percentage was less than 3% were labelled as “other”. It is worth noting that ICD codes for reimbursement purpose in Chinese herbal medicine might only indicate the location or the type of illness. Description of the ICD-9-CM/ICD-10-CM codes: A34, obstetrical tetanus; 460, acute nasopharyngitis (common cold); A46, erysipelas; A35, other tetanus; A31, infection due to other mycobacteria; 780, general symptoms; 784, symptoms involving head and neck; 536, disorders of function of stomach.

### 3.1. Descriptive Data

Table 1 summarizes the characteristics of the study participants. There were higher proportion of female (61.1% vs. 50.1%) in talc users than non-users, and the mean age of these two groups were 39.0 and 40.2 years old, respectively. There were no significant difference in monthly income, level of urbanization, and co-morbidity between talc users and non-users. During the follow-up from 1997 to 2013, the mean latency of talc-exposed patients and the diagnosis of stomach cancer was 4.3 years (standard error, 0.7 year). The mean duration of their talc exposure was 23.8 days (standard error, 10.7 days).

Table 2 shows the numbers of stomach cancers over time. Among people with age no less than 65 or male gender, there were higher incidence of stomach cancer (8.77 and 2.32 per 10,000 person-years) compared with their counterparts (1.31 and 1.42 per 10,000 person-years). For exposure of talc, there were higher incidence of stomach cancer in the talc-exposed period (2.49 per 10,000 person-years) than in the unexposed period (1.85 per 10,000 person-years). Classification by cumulative exposure of talc yielded incidence rates of stomach cancer of 2.13, 2.63, and 1.86 per 10,000 person-years among people with high (>21 g), medium (6–21 g), and low-to-none (≤6 g) exposure of talc, respectively.

### 3.2. Main Results

Table 3 shows the unadjusted and adjusted hazard ratios of stomach cancer by talc exposure. Subjects with exposure of talc had a higher hazard ratio of stomach cancer (adjusted ratio, 2.13; 95% CI, 1.54–2.94; *p* < 0.001). Classification by cumulative exposure of talc yielded adjusted hazard ratios of stomach cancer of 1.55 (95% CI, 0.79–3.17; *p* = 0.19) and 2.30 (95% CI, 1.48–3.57; *p* < 0.001) among subjects with high (>21 g) and medium (6–21 g) exposure of talc, as compared to the low-exposure counterparts. Confounding factors used for adjustment of hazard ratios included age, gender, and co-morbidity, as they were commonly considered to be relevant to stomach cancer [17]. The survival analysis showed that patients with talc exposure through Chinese herb medicine had a significant short cancer-free survival if compared with unexposed patients (*p* < 0.001) (Figure 3).

The cancer-free survival curve showed that patients with talc exposure through Chinese herb medicine had a significant short event-free survival (of stomach cancer) if compared with unexposed patients. For the survival plot a corresponding log-rank *p*-value was <0.001 (χ^2^ = 13.8; df = 1).

### 3.3. Other Analyses

Table 4 summarizes the sensitivity analyses for main results of the original model. Changing the cut-off point into the median of cumulative exposure in subjects who received talc prescription resulted in significantly increased hazard ratios in both subjects with high and low exposure of talc (1.89 and 3.01, respectively) as compared to the unexposed group, without showing a dose–response relationship, we also conducted a sensitivity analysis by changing the definition of high comorbidity from CCI > 2 into CCI > 0. This change did not affect the adjusted hazard ratios of stomach cancer by talc exposure. Applying a 5-year minimal induction period by excluding observations with time to event less than five years resulted in non-significantly increased hazard ratios in both subjects with high and medium exposure of talc (1.99 and 0.86, respectively) after checking the descriptive data.

## 4. Discussion

### 4.1. Key Results

We hypothesized that talc exposure would be associated with higher risk of stomach cancer, and that a positive dose-response effect exists. Our first hypothesis was confirmed. After adjustment for age, gender, and comorbidity, the data showed that people with talc exposure had a 2.1-fold increased risk of developing stomach cancer. However, our second hypothesis was not confirmed. The main results did not demonstrate a significant dose-response effect.

### 4.2. Mineral Analysis of Talc in Taiwan

Talc mine is usually interlaced with asbestos mine; therefore, talc is easily contaminated by asbestos or asbestiform fibers during the mining process [18]. TFDA prohibited the presence of asbestos in drugs and cosmetics since 2005 [19]. According to the Chinese Pharmacopoeia [20], suppliers of talcum powder for medical use should adopt one of the three qualified methods of examination to detect asbestos: (1)Fourier Transform Infrared spectrometry (FTIR)(2)X-ray Powder Diffraction (XRD)(3)Microscopy of asbestos

If the presence of asbestos was suspected in examination using the first or the second method, then the third method should be applied.

Due to a lack of publicly available report of mineral analysis of talcum powder in Taiwan, Song et al. examined 100 talc particles on 30 March 2018 using a scanning electron microscopy (SEM) with combination of energy dispersive X-ray spectroscopy (EDS) at 20,000× magnification to confirm the aspect ratio, length, diameter and structure according to the dust particles counting rule of ISO-14966 [21]. During the mineral analysis of talcum powder, no asbestos or asbestiform fibers were detected. The image files at 20,000× magnification are presented in the Appendix A. However, to our knowledge there was no information regarding the presence of asbestos in talcum powder for medical use during 1997–2005.

### 4.3. Limitations

Our study has some limitations that should be addressed. First, as a commonly discussed issue of the NHIRD, information about lifestyle and personal behavior, such as smoking, body-mass index and dietary factors, which might be associated with the risk of cancer, could not be extracted from the insurance database. Nevertheless, we had excluded patients with peptic ulcer disease or Helicobacter pylori infection prior to the inclusion date and made adjustment for comorbid conditions to attenuate the potential confounding effect.

Second, since ICD-9-CM codes were adopted to identify comorbidities and to calculate the CCI, we may overestimate the true prevalence, as physicians might make tentative diagnoses according to their impression. Although the over-diagnosis of comorbid conditions does not necessarily lead to a biased adjusted hazard ratio, it may result in underestimation of the confidence interval. The diagnosis of stomach cancer, on the other hand, was confirmed by the Registry for Catastrophic Illness Patients of which the application requires pathologic proof, and therefore not subject to the overestimation by ICD-9-CM codes.

Third, we did not included some risk factors of stomach cancer in our study, such as information of pernicious anemia, because in practice the ICD-9-CM codes related to these diagnoses were seldom used by physicians. In addition, the geographic location may also affect the risk of stomach cancer. Although we had compared the income and urbanization level between talc-exposed patients and the unexposed counterpart (Table 1), there were likely other environmental carcinogens associated with geographic locations that we were not able to adjust to the hazard ratios.

Fourth, information about talc exposure was based on prescription data of Chinese medicine doctor, and only talc powders in the form of extracted Chinese herbal products was reimbursed by the NHI. We could not establish information of people exposed to talc in the form of Chinese herbal decoction. The real exposure to talc might be underestimated. However, as discussed in a previous study by Chen et al. [22], because the NHI system has a coverage rate of 99.6% and the copayment is only about US$ 1.5 for each visit of a Chinese medicine doctor, which is much less than the cost of Chinese herbal decoction, we suggested that data from NHIRD still covers most of the talc exposure from Chinese herbal medicine, and would not confound our results.

### 4.4. Interpretation

Since our present paper studied talc as possible carcinogen, we referred to criteria proposed by Bradford Hill in 1965 as a guide for interpreting the results [23].

Effect size: The hazard ratio of stomach cancer for talc-exposed people was 2.13. It is lower than other known risk factor of stomach cancer, such as Helicobacter pylori infection, of which the odds ratio was 3.8 [24]. Nevertheless, a risk ratio greater than 2 is usually considered to be clinically relevant. It is worth noting that because the population exposed to talc was small, the confidence interval of the hazard ratio was relatively wide (1.54–2.94), and should be interpreted carefully. The rareness of the talc exposure was our reason to conduct this population-based cohort study with a large sample size to obtain reasonable estimates of risk.

Specificity and temporality: We treated talc exposure as time-dependent and only attributed stomach cancer occur afterward to the talc exposure, in order to ensure proper temporality. For the specificity, the talc exposure via Chinese herbal medicine was without asbestos, as regulated by Taiwan Food and Drug Administration (TFDA). We adjusted several major factors regarding stomach cancer, including age, gender, and comorbidities. However, some potential risk factors such as smoking, body-mass index and dietary factors were unable to be adjusted, as addressed in limitations.

Consistency: There is a paucity of research regarding the carcinogenic effects of talc ingestion on human gastrointestinal system. The IARC has classified talc containing asbestos as “carcinogenic to humans” (group 1), inhaled talc without asbestos as "not classifiable as to carcinogenicity in humans" (group 3), and perineal talc application as possibly carcinogenic to humans (group 2B); however, ingested talc has not yet been evaluated. Our recent systemic review and meta-analysis showed that workers with occupational talc exposure had an increased standardized mortality ratios of 1.21 (95% CI: 1.03–1.42, *p* = 0.02) for stomach cancer [25]. In the subgroup analysis, however, our previous meta-analysis showed that workers exposed to talc without asbestos had a non-significantly increased SMR of 1.26 (95% CI: 0.97–1.63, *p* = 0.09), probably due to insufficient case number. In the present study, we found an increased hazard ratio of 2.13 (95% CI: 1.54–2.94, *p* < 0.01) for stomach cancer in people received prescription of talc powder (without asbestos) from Chinese herb medicine. While showing consistent results, the risk estimates of these two studies could not be compared directly, because there was no quantification of talc ingestion for the workers included in the meta-analysis, whereas in this study the median of talc exposure was 10.5 g.

Biological gradient and plausibility: Regarding the dose-response effect, however, our study did not showed positive relationship between levels of talc exposure and the risk of stomach cancer. When using the first and the third quartile of cumulative talc exposure as cut-off points, the hazard ratio for stomach cancer for people with high level of talc exposure (>21 g) was not significantly increased, compared to the low-to-none group (≤6 g). In the sensitivity analysis, we found that when using the median of cumulative talc exposure as cut-off point, the hazard ratio for stomach cancer for people with high level of talc exposure (>10.5 g) was significantly increased, compared to the unexposed group (0 g). These findings may suggest a non-linear dose-response relationship for increased risk of stomach cancer by oral talc exposure. This inference becomes more reasonable if we take into consideration of possible mechanism for carcinogenicity of talc on gastrointestinal system. Talc powder was mostly not absorbable by human digestive organs, and the carcinogenic mechanism of talc was suspected to be chronic inflammation, based on limited animal models [26]. Another in vitro study by Davies et al. also found that talc is cytotoxic to macrophages, inducing fibrosis and chronic inflammation in animals [27]. Therefore, as ingestion of talc contributes to the inflammatory process of stomach, other environmental and genetic factors could affect the carcinogenic effect. Still, lack of dose-response relationship made increased risk of stomach cancer by talc less likely to be causal.

We also found that after excluding subjects with time to event less than five years there was no statistical significant correlation with the increased risk of stomach cancer. This finding could be related to insufficient number of cases after the exclusion, for there were only 13 cases of stomach cancer among patients with medium or high exposure to talc during the follow-up from 1997 to 2013 (Table 4). On the other hand, assuming chronic inflammation to be the mechanism for talc exposure causing stomach cancer, both exposure time and cumulative dose must be interacting with the genetic and environmental factors to determine the carcinogenicity. Therefore, stomach cancer might also occur in patients exposed to talc during a follow-up period less than five years.

### 4.5. Generalisability

Our database included a stratified randomly sampled Taiwanese cohort with population of 1,000,000. Since there was no known racial and ethnic difference in the carcinogenicity of talc or even asbestos, we assume that the results of this study are applicable to population worldwide. It is not suitable to apply our findings to pediatric patients since we only included adults in our study. The exposure of interest was oral intake of talc via Chinese herbal medicine; therefore, the risk estimates from our study could be comparable to future studies focusing on other sources of oral talc exposure without asbestos.

## 5. Conclusions

Based on our study, we concluded that there is positive association between increased risk of stomach cancer and oral intake of talc without asbestos. Although a linear dose-response relationship was not shown, we find it necessary to warn the Traditional Chinese physicians against the prescription of talc. Future research needs to focus on causal studies and to explore the possible mechanisms of carcinogenicity of talc in human gastrointestinal system.

## Figures and Tables

**Figure 1 ijerph-16-00717-f001:**
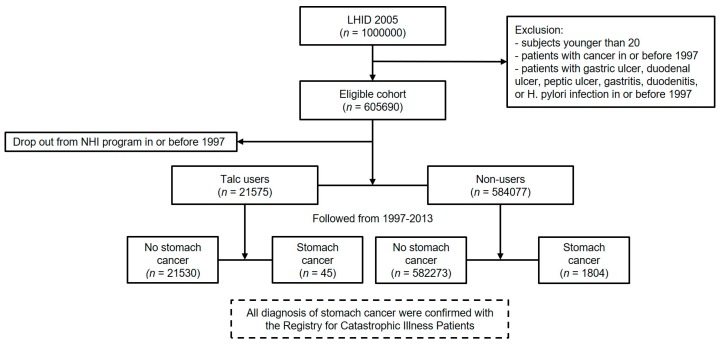
Data collection diagram.

**Figure 2 ijerph-16-00717-f002:**
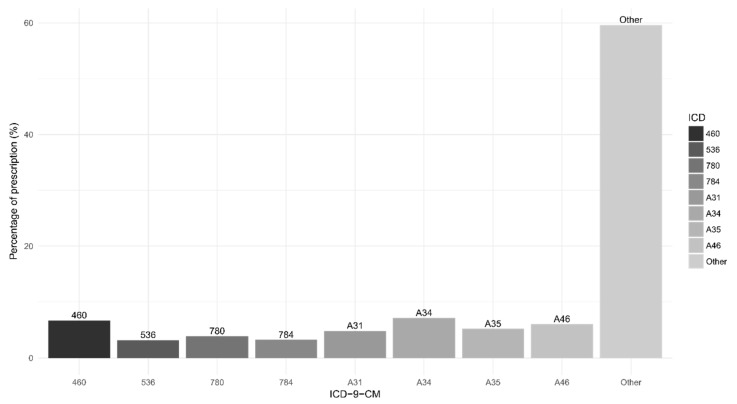
Main diagnosis for prescription of talcum powder.

**Figure 3 ijerph-16-00717-f003:**
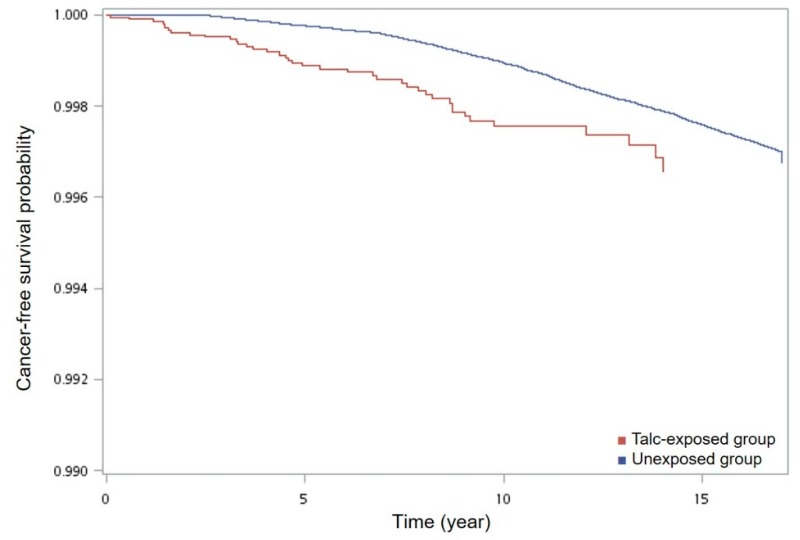
Cancer-free survival curve of patients with and without talc ingestion. Bottom half of figure is meaningless.

**Table 1 ijerph-16-00717-t001:** Characteristics of study base at enrollment.

Variable	Non-Users *N* = 584,077	Talc Users *N* = 21,575
Gender		
Female	292,319 (50.1%)	13,180 (61.1%)
Male	291,758 (49.9%)	8395 (38.9%)
Age (year) ^a^	40.2 ± 14.7	39.0 ± 13.2
Monthly Income ^b^	21,822 ± 12,575	21,630 ± 11,380
Urbanization ^c^		
Level 1 (high)	173,513 (29.9%)	5,486 (25.1%)
Level 2 (medium)	249,744 (43.0%)	9602 (44.7%)
Level 3 (low)	157,831 (27.2%)	6414 (29.8%)
Charlson Comorbidity Index ^d^		
0–2	583,744 (99.94%)	21,558 (99.92%)
>2	333 (0.06%)	17 (0.08%)
Follow-up time in month	196.2 ± 21.7	200.3 ± 13.4

^a^ Age at the beginning of the study period (1 January 1997); ^b^ Counted in New Taiwan Dollar (TWD); ^c^ Number of missing = 3062; ^d^ The Charlson Comorbidity index was calculated with exclusion of cancer, as these patients were already excluded from the study.

**Table 2 ijerph-16-00717-t002:** Numbers of stomach cancer over time by subgroups.

Variable	Person-Years	No. of Stomach Cancer	Rate/10,000 Person-Years (95% CI ^a^)
Age			
<65	9,171,234	1202	1.31 (1.24–1.39)
≥65	737,872	647	8.77 (8.11–9.47)
Gender			
Female	5,024,262	715	1.42 (1.32–1.53)
Male	4,884,844	1134	2.32 (2.19–2.46)
Urbanization			
Level 1 (high)	2,944,130	519	1.76 (1.61–1.92)
Level 2 (medium)	4,247,016	796	1.87 (1.75–2.01)
Level 3 (low)	2,668,495	516	1.93 (1.77–2.11)
Charlson Comorbidity Index ^b^			
0–2	9,904,066	1848	1.87 (1.78–1.95)
>2	5040	1	1.98 (0.05–11.06)
Exposure of talc ^c^			
Unexposed period	9,728,639	1804	1.85 (1.77–1.94)
Talc-exposed period	180,467	45	2.49 (1.82–3.34)
Cumulative talc exposure ^c^			
Low to none (≤ 6 g)	9,774,552	1816	1.86 (1.77–1.95)
Medium (6–21 g)	87,550	23	2.63 (1.67–3.94)
High (>21 g)	47,004	10	2.13 (1.02–3.91)

^a^ The 95% confidence interval of incidence rate were estimated by the Fisher’s exact test; ^b^ The Charlson Comorbidity Index (CCI) was calculated with exclusion of malignancies; ^c^ Summary statistics of exposure: median = 10.5 g, first quartile = 6 g, third quartile = 21 g. The exposure of talc was treated as a time-dependent variable.

**Table 3 ijerph-16-00717-t003:** Hazard ratio of stomach cancer by talc exposure.

Risk Factor	Person-Years (Cases)	Crude Hazard Ratio ^a^		Adjusted Hazard Ratio ^b^		Ten-Year Absolute Risk/1000 Persons (95% CI)
		HR (95% CI)	*p*-Value	HR (95% CI)	*p*-Value	
Exposure of talc						
Unexposed period	9,728,639 (1804)	1.00		1.00		1.07 (0.99–1.15)
Talc-exposed period	180,467 (45)	**1.83 (1.32–2.52)**	**<0.001**	**2.13 (1.54–2.94)**	**<0.001**	**1.95 (1.33–2.57)**
Cumulative talc exposure						
Low to none (≤6 g)	9,774,552 (1816)	1.00		1.00		1.07 (0.99–1.16)
Medium (6–21 g)	87,550 (23)	**1.99 (1.28–3.09)**	**0.002**	**2.30 (1.48–3.57)**	**<0.001**	**2.13 (1.19–3.07)**
High (>21 g)	47,004 (10)	1.43 (0.71–2.87)	0.31	1.58 (0.79–3.17)	0.19	1.54 (0.47–2.60)

Bold numbers: significant results. HR, hazard ratio. ^a^ The hazard ratios were estimated using the Cox proportional hazard model, with talc exposure treated as a time-dependent variable. ^b^ The adjusted hazard ratio was adjusted by age, gender, and Charlson Comorbidity Index (CCI) excluding malignancies.

**Table 4 ijerph-16-00717-t004:** Sensitivity analyses of talc exposure on the risk of stomach cancer.

Model	Person-Years (Cases)	Adjusted Hazard Ratio	
		HR (95% CI)	*p*-Value
Original model			
Cumulative talc exposure			
Low to none (≤ 6g)	9,774,552 (1816)	1.00	
Medium (6~21g)	87,550 (23)	2.30 (1.48–3.57)	<0.001
High (>21g)	47,004 (10)	1.58 (0.79–3.17)	0.19
Changing the cut-off points for levels of talc exposure ^a^			
Cumulative talc exposure			
Unexposed	9,728,639 (1804)	1.00	
Low (≤10.5 g)	88,312 (23)	2.40 (1.54–3.73)	0.007
High (>10.5 g)	92,155 (22)	1.89 (1.19–3.01)	<0.001
Excluding time to event less than five years ^b^			
Cumulative talc exposure			
Low to none (≤6 g)	9,755,288 (1671)	1.00	
Medium (6–21 g)	78,828 (6)	0.86 (0.39–1.93)	0.72
High (>21 g)	43,288 (7)	1.99 (0.81–3.59)	0.16

The adjusted hazard ratio was adjusted by age, gender, and Charlson Comorbidity Index (CCI) excluding malignancies. Bold numbers: significant results. ^a^ The cut-off point for distinguishing levels of talc exposure was changed into the median of cumulative exposure in subjects received talc prescription. ^b^ The time to event (stomach cancer, drop-out, or follow-up endpoint) less than five years was excluded to ensure the 5-year minimal induction period of talc to stomach cancer.

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
