# Peer review of "Stomach Cancer and Exposure to Talc Powder without Asbestos via Chinese Herbal Medicine: A Population-Based Cohort Study"

_ijerph, 2019, doi:10.3390/ijerph16050717_

Round 1
Reviewer 1 Report
The paper deals with the suspected carcinogenicity of talc powder containing in Chinese herbal medicine preparations. The paper follows a meta-analysis of the same group which concluded that " Overall, our study suggested that occupational talc (contaminated with asbestos) exposure is positively related to an increased risk of stomach cancer. However, the evidence is not sufficiently robust to determine the association between exposure to talc not containing asbestiform fibers and the risk of stomach cancer." Using adequate statistical tools the authors conclude that there is a positive association between increase risk of stomach cancer and oral intake of talc without asbestos. I have some concerns about "Materials and Methods and Discussion session.
Material and Methods:
The authors do not describe the medical preparation prescribed. According to TFDA this is asbestos free (and also asbestiform-fibre free?). However it could be important to report how this was proved (microscopy or other).
Is the same preparation used by all the subjects? Or different preparations are employed?
In my opinion also the reason for the prescription of these preparation/s should be at least more widely described, if possible, in the Introduction.
Result and Discussion
The authors report that excluding subjects exposed for less than 5 years no statistical significant correlation was found with the increased risk of stomach cancer. However this finding is not discussed. In my opinion it could be related also to the absence of dose response relationship (lasting of chronic inflammation).
Finally, in my opinion and if possible, considering pros and cons, the authors in the Conclusion, should discourage the use of talc as therapeutic tool or at least alert against its use.
Author Response
Point-by-point response to the comments of reviewer 1.
Point 1.
According to TFDA this (talc) is asbestos free (and also asbestiform-fibre free?). However, it could be important to report how this was proved (microscopy or other).
Response:
Thank you for this important suggestion. TFDA had prohibited the presence of asbestos (without mentioning asbestiform fibers) in drugs and cosmetics since 2005 [1]. According to the Chinese Pharmacopoeia [2], suppliers of talcum powder for medical use should adopt one of the three qualified methods of examination to detect asbestos:
1. Fourier Transform Infrared spectrometry (FTIR)
2. X-ray Powder Diffraction (XRD)
3. Microscopy of asbestos
If the presence of asbestos was suspected in examination using the first or the second method, then the third method should be applied.
In addition, we also performed a mineral analysis of talcum powder in Taiwan following ISO-14966 [3]. There were no asbestos or asbestiform fibers detected, and we presented the image files at 20,000x magnification in the supplement. However, to our knowledge there was no information regarding the presence of asbestos in talcum powder for medical use during 1997-2005.
We have revised the discussion section by adding the following:
4.2. Mineral analysis of talc in Taiwan
Talc mine is usually interlaced with asbestos mine; therefore, talc is easily contaminated by asbestos or asbestiform fibers during the mining process [18]. TFDA prohibited the presence of asbestos in drugs and cosmetics since 2005 [19]. According to the Chinese Pharmacopoeia [20], suppliers of talcum powder for medical use should adopt one of the three qualified methods of examination to detect asbestos:
1. Fourier Transform Infrared spectrometry (FTIR)
2. X-ray Powder Diffraction (XRD)
3. Microscopy of asbestos
If the presence of asbestos was suspected in examination using the first or the second method, then the third method should be applied.
For lack of publicly available report of mineral analysis of talcum powder in Taiwan, S.-R. Song, Y.-C. Lu, and H.-Y. Peng examined 100 talc particles using a scanning electron microscopy (SEM) with combination of energy dispersive x-ray spectroscopy (EDS) at 20,000x magnification to confirm the aspect ratio, length, diameter and structure according to the dust particles counting rule of ISO-14966 [21]. During the mineral analysis of talcum powder, no asbestos or asbestiform fibers were detected. The image files at 20,000x magnification were presented in the supplement (Appendix 5). However, to our knowledge there was no information regarding the presence of asbestos in talcum powder for medical use during 1997-2005.
Reference:
1. TFDA Regulation of Drugs No. 0940338432. Availabe online: https://www.fda.gov.tw/tc/includes/GetFile.ashx?id=f636694164952134262 (accessed on February 9th 2019).
2. zhong hua yao dian [Chinese Pharmacopoeia], 8th ed. (page 2354); TFDA: 2016.
3. ISO. Ambient air–determination of numerical concentration of inorganic fibrous particles, scanning electron microscopy method. International Organization for Standardization, Method ISO 14966: 2002.
Point 2.
The authors do not describe the medical preparation (of talc) prescribed. Is it the same preparation used by all the subjects? Or different preparations are employed?
Response:
Thank you for raising this issue.
In our study, only exposure to talc powder in the form of extracted Chinese herbal products was included, as one limitation of accessing data from the reimbursement database (“4.2. Limitations”). The other form of preparation is the traditional Chinese herbal decoction, which is not reimbursed by Taiwan’s National Health Insurance Administration.
The real exposure to talc might therefore be underestimated. However, as discussed in a previous study by Chen et al., because the NHI system has a coverage rate of 99.6% and the copayment is only about US$ 1.5 for each visit to a Chinese medicine doctor, which is much less than the cost of Chinese herbal decoction, we suggested that our data still covers most of the talc exposure from Chinese herbal medicine, and would not confound our results.
Accordingly, we have revised the first paragraph of corresponding session “2.4. Exposure Assessment” by adding following description:
“NHI of Taiwan only reimburses prescription of talc powder in the form of extracted Chinese herbal products; Therefore, talc in the form of traditional Chinese herbal decoction was not included in the database.”
Reference:
- Chen, M.-C.; Lai, J.-N.; Chen, P.-C.; Wang, J.-D. Concurrent use of conventional drugs with Chinese herbal products in Taiwan: a population-based study. Journal of traditional and complementary medicine 2013, 3, 256-262.
Point 3.
In my opinion also the reason for the prescription of these preparation/s should be at least more widely described, if possible, in the Introduction.
Response:
Thank you for this important suggestion. In traditional Chinese medicine, talc is adopted as antipyretics and diuretics[1] (“Introduction”). However, talc can also be prescribed with other herbal medicine in various conditions. Therefore, we also described the diagnoses for prescription of talc powder and showed the percentages in the bar chart (Figure 3). As shown in our data, talc was prescribed mainly in infectious diseases for symptom relief.
We have revised the manuscript by adding this figure with related description after the first paragraph of the result section.
Figure 2. Main diagnosis for prescription of talc powder
The bar chart showed the ICD codes and the percentages of all prescription of talc powder. ICD codes of which percentage was less than 3% were labelled as “other”. It is worth noting that ICD codes for reimbursement purpose in Chinese herbal medicine might only indicate the location or the type of illness. Description of the ICD-9-CM/ICD-10-CM codes: A34, obstetrical tetanus; 460, acute nasopharyngitis (common cold); A46, erysipelas; A35, other tetanus; A31, infection due to other mycobacteria; 780, general symptoms; 784, symptoms involving head and neck; 536, disorders of function of stomach.
Reference:
- Li, S.Z. ben cao gang mu [Compendium of Materia Medica]. Shanghai Ancient Books Publishing House 1991, 9.
Point 4.
The authors report that excluding subjects exposed for less than 5 years no statistical significant correlation was found with the increased risk of stomach cancer. However this finding is not discussed. In my opinion it could be related also to the absence of dose response relationship (lasting of chronic inflammation).
Response:
Thank you for this illuminating comment. We have made the revision by adding the following paragraph to the end of “4.4. interpretation”.
We also found that after excluding subjects with time to event less than five years there was no statistical significant correlation with the increased risk of stomach cancer. This finding could be related to insufficient number of cases after the exclusion, for there were only 13 cases of stomach cancer among patients with medium or high exposure to talc during the follow-up from 1997 to 2013 (Table 4). On the other hand, assuming chronic inflammation to be the mechanism for talc exposure causing stomach cancer, both exposure time and cumulative dose must be interacting with the genetic and environmental factors to determine the carcinogenicity. Therefore, stomach cancer might also occur in patients exposed to talc during a follow-up period less than five years.
Point 5.
Finally, in my opinion and if possible, considering pros and cons, the authors in the Conclusion, should discourage the use of talc as therapeutic tool or at least alert against its use.
Response:
Thank you for the suggestion. We have revised our conclusions into the following paragraph:
Based on our study, we concluded that there is positive association between increased risk of stomach cancer and oral intake of talc without asbestos. Although a linear dose-response relationship was not shown, we find it necessary to warn the Traditional Chinese physicians against the prescription of talc. Future research needs to focus on causal studies and to explore the possible mechanisms of carcinogenicity of talc in human gastrointestinal system.

Reviewer 2 Report
The authors did a population based study and demonstrated that Talc exposure through traditional Chinese herbal medicine is linked to stomach cancer. The paper is well-written and brought an important topic regarding the carcinogenesis of Talc. However, there are concerns and questions that need to be clarified in order to draw a conclusion.
1. The Authors must provide evidence to convince the reader that the talc is free of asbestos as claimed. There has to be an independent and reliable evaluation of the powder by mineralogists that certifies that this talc power present in those Chinese herbal medicine is asbestos free.
2. The Authors need to provide information about duration of exposure and latency (i.e., hom many years after initial exposure did the patients get gastric cancer?). For example regardless of amount of exposure, ~99% of mesothelioma occurs only 30+ years from initial exposure. If a similar latency applies to talc it could explain the absence of dose response.
3. The authors need to consider the geographic locations of the cancer patients to exclude the possibility that there is other cause that might lead to stomach cancer besides Talc exposure, such as environmental exposure to some carcinogen.
4. How did the authors choose the amounts to set for the levels of exposure of Talc between high (>21g), medium (6-21g) and low (<6g)? These levels are based on what? Is there any previous study or publication to support the settings?
5. The number of individuals in none-Talc-users group is about 27 times comparing to Talc users group. The authors need to take into consideration whether the very unbalanced numbers may influence the results.
Author Response
ijerph-440170 Point-by-point response to the comments of reviewer 2.
Point 1.
The Authors must provide evidence to convince the reader that the talc is free of asbestos as claimed. There has to be an independent and reliable evaluation of the powder by mineralogists that certifies that this talc power present in those Chinese herbal medicine is asbestos free.
Response:
Thank you for raising this issue. TFDA had prohibited the presence of asbestos in drugs and cosmetics since 2005 [1]. According to the Chinese Pharmacopoeia [2], suppliers of talcum powder for medical use should adopt one of the three qualified methods of examination to detect asbestos:
1. Fourier Transform Infrared spectrometry (FTIR)
2. X-ray Powder Diffraction (XRD)
3. Microscopy of asbestos
If the presence of asbestos was suspected in examination using the first or the second method, then the third method should be applied.
We found that currently there was no publicly available report of mineral analysis of talcum powder in Taiwan. Therefore, we performed a mineral analysis of talcum powder in Taiwan following ISO-14966 [3]. There were no asbestos or asbestiform fibers detected, and we presented the image files at 20,000x magnification in the supplement. However, to our knowledge there was no information regarding the presence of asbestos in talcum powder for medical use during 1997-2005.
We have revised the discussion section by adding the following:
4.2. Mineral analysis of talc in Taiwan
Talc mine is usually interlaced with asbestos mine; therefore, talc is easily contaminated by asbestos or asbestiform fibers during the mining process [18]. TFDA prohibited the presence of asbestos in drugs and cosmetics since 2005 [19]. According to the Chinese Pharmacopoeia [20], suppliers of talcum powder for medical use should adopt one of the three qualified methods of examination to detect asbestos:
1. Fourier Transform Infrared spectrometry (FTIR)
2. X-ray Powder Diffraction (XRD)
3. Microscopy of asbestos
If the presence of asbestos was suspected in examination using the first or the second method, then the third method should be applied.
For lack of publicly available report of mineral analysis of talcum powder in Taiwan, S.-R. Song, Y.-C. Lu, and H.-Y. Peng examined 100 talc particles using a scanning electron microscopy (SEM) with combination of energy dispersive x-ray spectroscopy (EDS) at 20,000x magnification to confirm the aspect ratio, length, diameter and structure according to the dust particles counting rule of ISO-14966 [21]. During the mineral analysis of talcum powder, no asbestos or asbestiform fibers were detected. The image files at 20,000x magnification were presented in the supplement (Appendix 5). However, to our knowledge there was no information regarding the presence of asbestos in talcum powder for medical use during 1997-2005.
Reference:
1. TFDA Regulation of Drugs No. 0940338432. Availabe online: https://www.fda.gov.tw/tc/includes/GetFile.ashx?id=f636694164952134262 (accessed on February 9th 2019).
2. zhong hua yao dian [Chinese Pharmacopoeia], 8th ed. (page 2354); TFDA: 2016.
3. ISO. Ambient air–determination of numerical concentration of inorganic fibrous particles, scanning electron microscopy method. International Organization for Standardization, Method ISO 14966: 2002.
Point 2.
The Authors need to provide information about duration of exposure and latency (i.e., how many years after initial exposure did the patients get gastric cancer?). For example regardless of amount of exposure, ~99% of mesothelioma occurs only 30+ years from initial exposure. If a similar latency applies to talc it could explain the absence of dose response.
Response:
Thank you for this enlightening comment. During the follow-up from 1997 to 2013 in our data, the mean latency of talc-exposed patients and the diagnosis of stomach cancer was 4.3 years (standard error, 0.7 year). The mean duration of their talc exposure was 23.8 days (standard error, 10.7 days). The reason of the shorter latency as shown in our study could result from the limitation of our follow-up period. Further analysis of the same cohort with longer time of observation (e.g. from 1997 to 2019) may provide stronger evidence of the carcinogenicity of orally ingested talc.
Accordingly, we have revised our manuscript by adding the information above to our result (“3.1. Descriptive data”).
Point 3.
The authors need to consider the geographic locations of the cancer patients to exclude the possibility that there is other cause that might lead to stomach cancer besides Talc exposure, such as environmental exposure to some carcinogen.
Response:
Thank you for this valuable suggestion. According to the Cancer Registry Annual Report of Taiwan in 2013, the age-adjusted incidence rate of stomach cancer for men was 16.04, 12.01, 13.52, and 15.00 in Northern, Middle, Southern, and Eastern Taiwan, respectively. For women the age-adjusted incidence rate was 8.69, 6.12, 6.71, and 8.40. The geographic location is reasonably related to cancer incidence, and we consider the main mediating factors to be the social-economic status, such as income and urbanization level. Therefore, we compared the income and urbanization level between talc-exposed patients and the unexposed counterpart (Table 1). We did not put these two variables into the regression model because the difference was small.
Nevertheless, there were likely other environmental carcinogens associated with geographic locations that we were not able to adjust to the hazard ratios. We have revised the manuscript by adding it to the discussion (“4.3. Limitations”).
Reference:
- Cancer Registry Annual Report, Taiwan: Health Promotion Administration (Taiwan); 2013 [Available from: https://www.hpa.gov.tw/File/Attach/5191/File_6166.pdf, page 197]
Point 4.
How did the authors choose the amounts to set for the levels of exposure of Talc between high (>21g), medium (6-21g) and low (<6g)? These levels are based on what? Is there any previous study or publication to support the settings?
Response:
Thank you for this reminding comment. The absolute numbers for categorizing levels of talc exposure were not predetermined arbitrarily, because we did not find related evidence in previous studies; instead, we used the first and the third quartile of cumulative dose in subjects with talc intake as cut-off points to categorize the exposure into three levels (≤ first quartile or unexposed, first quartile ~ third quartile, > third quartile). We mentioned this setting in the “2.5. Statistical Analysis” when describing the sensitivity analysis, as we also applied the median value of the cumulative talc exposure as new cut-off point, in order to make three levels of talc exposure (unexposed, ≤ median, > median) to minimize the impact of arbitrariness. In table 2, we also reported the summary statistics of talc exposure in the table footnotes (median = 10.5g, first quartile = 6g, third quartile = 21g).
However, we should have properly describe the setting of cut-off points more explicitly in the “2.3. Variables”. Accordingly, we have revised this section by adding the following description in the end of the paragraph.
“Regarding the talc exposure, we used the first and the third quartile of cumulative dose in subjects with talc intake as cut-off points to categorize the exposure into three levels.”
Point 5.
The number of individuals in none-Talc-users group is about 27 times comparing to Talc users group. The authors need to take into consideration whether the very unbalanced numbers may influence the results.
Response:
Thank you for the comment. Because there were not many patients that took talc as Chinese herbal medicine, the exposure to talc could be consider as rare exposure / rare treatment. To study the relationship between a relatively rare disease (stomach cancer) and a relatively rare exposure (talcum powder) require a large sample size to obtain reasonable estimates of risk. Therefore, we conducted a population-based cohort study to approach this topic.
We have revised the second paragraph of “4.4. Interpretation” to properly address this issue.
“Effect size: The hazard ratio of stomach cancer for talc-exposed people was 2.13. It is lower than other known risk factor of stomach cancer, such as Helicobacter pylori infection, of which the odds ratio was 3.8 [20]. Nevertheless, a risk ratio greater than 2 is usually considered to be clinically relevant. It is worth noting that because the population exposed to talc was small, the confidence interval of the hazard ratio was relatively wide (1.54-2.94), and should be interpreted carefully. The rareness of the talc exposure was our reason to conduct this population-based cohort study with a large sample size to obtain reasonable estimates of risk.”

Round 2
Reviewer 2 Report
The authors addressed the questions carefully, and they added additional data to support their findings.